# "We Are Just Supposed to Be an NGO Helping": A Qualitative Case Study of Health Workers' and Volunteers' Perceptions of the Government and Civil Society's Role in Fighting Jiggers in Bungoma County, Kenya

Åse Walle Mørkve [1,*], Jackline Sitienei [2] and Graziella Van den Bergh [3]

1 Department of Intercultural Studies, NLA University College, 5812 Bergen, Norway
2 Department of Health Policy Management and Human Nutrition, Moi University, Eldoret 30100, Kenya; jcsitienei@gmail.com
3 Department of Health and Functioning, Section for Global Health and Rehabilitation, Western Norway University of Applied Sciences, 5020 Bergen, Norway; gvb@hvl.no
* Correspondence: asemor@nla.no

**Abstract:** Non-governmental organizations (national and international) are important actors in addressing health issues in Kenya. Sandflea/jigger infections (tungiasis) are a public health challenge that severely affect children, older adults, and other vulnerable people in poor communities worldwide. In Kenya, NGOs have been involved in sandflea eradication for more than twenty years. Without treatment, the flea may cause debilitating infections and sores, resulting in difficulties with walking and grasping, as well as social harassment. This paper aims to shed light on health workers' and volunteers' perceptions of the government and civil society's role in fighting jigger infections. Data were collected through a qualitative case study design, with a three-month fieldwork including participation in mobile jigger removal programs, 18 semi-structured in-depth interviews, informal talks, and observations, in five villages in Bungoma County. The thematic analysis of the data resulted in three recurring themes: (1) the NGO-driven jigger program as a (fragile) resource for local communities, (2) the need for more consistent collaboration between NGOs and public health services, and (3) the local perceptions of the governments' responsibilities in combatting the plague. The findings imply that the 10-year-old national policy guidelines on the prevention and control of jigger infestations need to be updated; this includes the coordination of the public and private actors' roles, the incorporation of lessons learned, and the need for a multisectoral One Health approach to combat the jigger menace in the country.

**Keywords:** sandfleas; jigger; tungiasis; non-governmental organizations; Kenya; public health; health promotion; mobile clinics



## 1. Introduction

The health sector can be described as being composed of three parts: the private for-profit sector, the public sector, and civil society, often referred to as the "third sector" or the "non-profit sector" [1]. The civil society sector often involves voluntary work towards a collective action, which, at a larger, organized scale, is referred to as non-governmental organizations (NGOs). NGOs have over the last thirty years become important actors in the health sector and have gained considerable influence in governments and societies worldwide [1]. Even though local, national, or international NGOs may operate relatively independently of the government, there is often some degree of governmental control when these are established or allowed to operate. In Kenya, where this study was conducted, it is estimated that more than 11,000 NGOs are operating all over the country in different sectors, dealing with health, human rights, environmental issues, and advocacy [2,3]. The

Ministry of State within the Office of the President has a board that is responsible for regulating and enabling the NGO sector of Kenya [4].

Sandflea disease, or a jigger infection as it is called in popular terms, or tungiasis as it is called in medical terms [5], is a public health issue that several NGOs have been working on in Kenya for the last twenty years [6,7]. This parasitic skin infection is caused by the flea Tunga Penetrans, and is found in various types of soils, with dry and sandy ground being particularly suited for their development [8]. The jigger flea penetrates the skin of the host (Figure 1), both human or animal, mainly on body parts that are in contact with the soil [9]. Children, people with disabilities, and older people are most vulnerable, especially when more frequently in contact with infested surroundings and dirt. If not properly removed, the flea stays in the skin for almost six weeks. The penetration site becomes irritated and very itchy [9]. Without proper treatment, which is often not available in poor, rural communities, the flea may cause severe illnesses, such as bacterial infections, deep fissures, ulcers, the loss of nails, edema, suppuration [8,10], tetanus, if not vaccinated, and even gangrene [8]. Those affected might have difficulties walking and grasping [11], thus hindering everyday activities. Stigmatization and social harassment, such as bullying and isolation, have also been reported, resulting in many children dropping out of school [6,10].

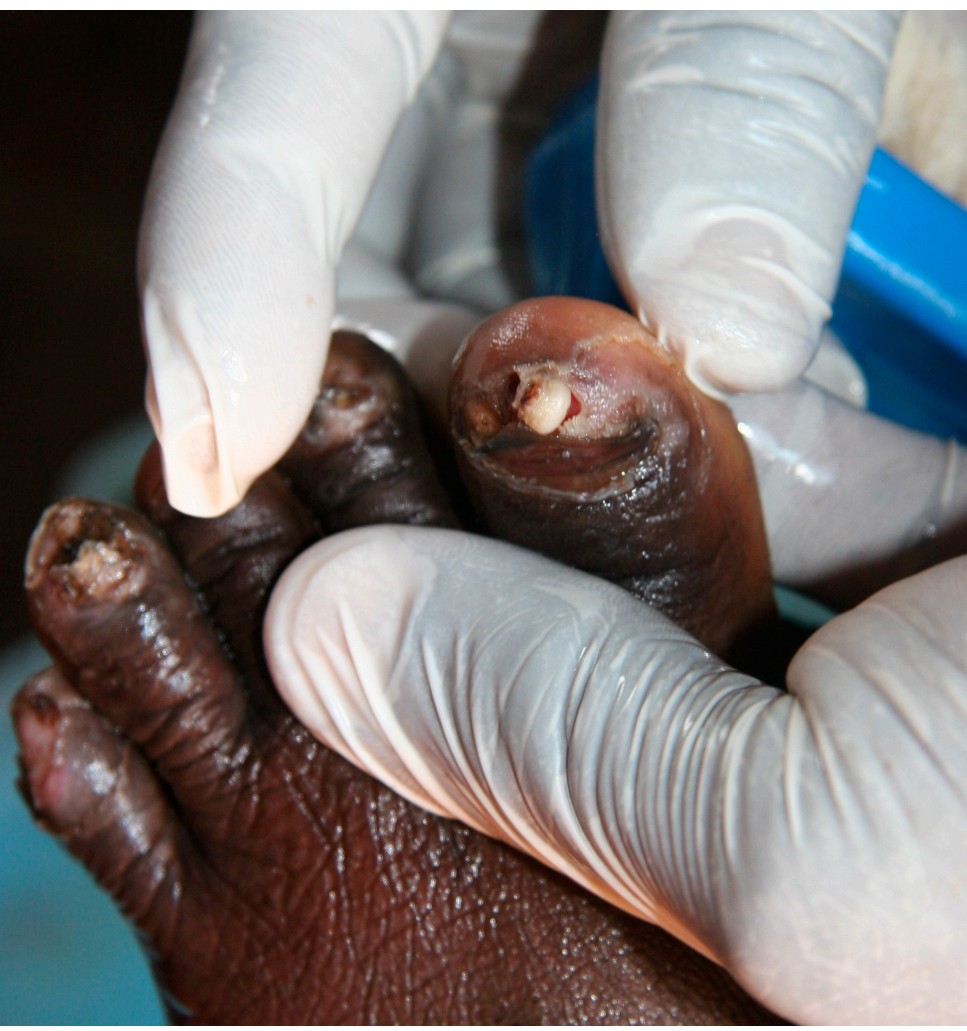

**Figure 1.** Jigger flea sores in a child's feet. Picture taken with permission by first author.

Globally, jiggers mainly affect poorer communities in Africa, Central and South America, and India. It is estimated that hundreds of millions of people in almost 90 countries are at risk of this parasitic infection [12], and a jigger infection is defined by the World

Health Organization as a neglected tropical disease [13], whose burden is not sufficiently recognized by the scientific community, the health sector, or policymakers [5]. To raise awareness of this plague, effective health promotion measures are needed at the national level [14]. In Kenya, it is estimated that 10 million people are at risk of contracting a jigger infection [15]. In Central Kenya, where jiggers are hyperendemic, a prevalence of as much as 57% has been recorded among children [16].

The Department of Environmental Health and Sanitation and the Vector and Vermin Control Unit within the Ministry of Public Health and Sanitation has attempted to fight jigger infections [7]. In 2014, the Ministry of Health's Division of Environmental Health published national policy guidelines on the prevention and control of jigger infestation, recognizing that jiggers are an important yet neglected public health problem in Kenya. According to the guidelines, NGOs have strongly supported the ministry in factors such as awareness creation, advocacy, treatment, and control since 2004. The guidelines emphasize the need to focus on prevention and control; for example, in schools and outreach camps, including information on environmental and personal hygiene, the chemical control of jigger fleas, and the use of repellant as treatment [7].

According to the World Health Organization, a multisectoral One Health approach is needed for the sustainable control of jigger fleas and infections within highly endemic communities. The multisectoral One Health approach involves collaboration and coordination between different sectors, such as environmental health, human health, and animal health in order to address health issues at local, national, and global levels [17,18]. Through a case study design including various data collection methods over time, supplemented by more recent situation reports [14], this study aims to understand the role division between the public and private not-for-profit/NGO sector in fighting this public health challenge in Bungoma County. This will be achieved by analyzing community-level perspectives among the volunteers and staff working in these NGOs in conjunction with the perspectives of governmental health workers, health service providers, and public health officers. These local experiences of the jigger plague will be discussed in the light of decades of NGO contributions towards jigger eradication. Moreover, the current study aims to fill the gap in qualitative research on the jigger menace in Kenya [5,12], and contribute to an understanding of the effectiveness and sustainability of applied health promotion programs, public policies, and guidelines on jigger infections in order to reach health and well-being for all [14,18,19].

This study is inspired by the health promotion theory and the Ottawa Charter (1986), which both emphasize advocacy, enablement, and mediation through collaboration across all sectors, allowing all people to achieve health equity. Health promotion indicates five key action areas: to build healthy public policies, create supportive environments for health, strengthen community action for health, develop personal skills, and re-orient health services [20]. The study is also inspired by the Community Coalition Action Theory (CCAT). This theory seeks to explain how organizational structure, function, and effectiveness across organizations may be improved and may benefit all the partners within the community, working together to change policies towards a healthier living for the community. The theory consists of several constructs such as lead agencies, member engagement, assessment and planning, community capacity, and the development of policies [21].

In this case study, the main research questions are as follows: What is the perceived role division between the government and civil society organizations according to those fighting jigger infections at the grassroots level, and what are the opportunities and challenges encountered? How do local experiences and perspectives resonate with the last decade's national health promotion policies and guidelines on jigger infections and control in Kenya?

## 2. Materials and Methods

### 2.1. Study Design

In this qualitative case study [14], the first author carried out three months of fieldwork, while participating in an NGO-driven mobile jigger removal program conducted by the

Bungoma Red Cross in Bungoma County. An earlier paper has addressed the experience of those affected by jigger infections: how they suffer from the condition, their own explanations of the causes of the infection, and the emergence of hopelessness and fatalism due to the reoccurrence of the jigger fleas after treatment [6]. In turn, this paper aims to shed light on health workers' and volunteers' perceptions of the governments and civil society's role in fighting jigger infections. NGOs fighting jigger fleas have been operating in highly affected areas of Bungoma since 2002 [6]. We address the perspectives of staff and volunteers from NGOs, as well as the perspectives of community health workers and public health officers in Bungoma County, on the functioning of the programs that aim to fight sandflea infestations, and this is subsequently addressed by local and public stakeholders at differing levels.

### 2.2. Study Context

The study was carried out in Bungoma County, situated in the western part of Kenya. Bungoma County is the second largest county in Kenya, with a total population of almost 1.7 million inhabitants. A majority of the population (88%) live in rural areas. About 44% of the population (approximately seven hundred thousand people) are children between 0–14 years [22,23]. Agriculture is the most important economic activity. Animal husbandry and pastoralism, which both hold implications for fighting jigger infections as animals are common reservoirs for jigger fleas, are prominent features of the region. Bungoma County has a total of 184 health facilities: 17 hospitals, 14 health centers, 102 dispensaries, 20 faith-based organizations, and 52 private clinics. In Kenya, the health care system is organized into six levels of care, the first being the community level. At the community level, the community health worker (primarily based on voluntary work) oversees a certain number of households within the village. Level two and level three services are provided at the dispensaries and health centers, respectively. Levels one, two, and three are categorized as primary healthcare services, and deal mainly with health promotion, disease prevention, and essential service provision. Levels four, five, and six are the county hospitals, the county referral hospitals, and the national referral hospitals, respectively [24].

During this study, the NGO-driven removal clinic visited five rural villages in the county, including Ndivisi, Chwele, Bumula, Kanduyi, and Nalondo. All villages are situated in rural areas, counting about 2–3000 inhabitants each [6]. The program aimed to treat people suffering from jigger infections, as well as to raise awareness in the community. The Bungoma Red Cross program employed one NGO staff member and four-to-five NGO volunteers, who traveled around once a week, or more when funding allowed it, to sandflea-infested villages in the county. Volunteers who worked in the removal program were remunerated by the NGO with KES 300 per working day. Typically, the NGO staff identified high-prevalence areas by talking to local community health workers in the divisions. The local community health worker typically works on the ground, reporting back to the public health officer in the community. The community health worker was asked to mobilize and inform those affected about the upcoming jigger removal program in their respective division, one week prior to the jigger removal day. After one week, the NGO staff and volunteers came to the identified place, bringing different drugs and equipment to treat those coming to the clinics (Figure 2). In addition, during the jigger removal program, the NGO workers talked to each participant to raise awareness of jigger prevention measures.

### 2.3. Data Collection and Analysis

During the three-month fieldwork period, the first author collected data through unstructured observations, informal talks, and fieldnotes, while participating in five jigger removal clinics. There were a total of 55 participants in the study, whereof 18 semi-structured in-depth interviews are included in this current article, including volunteers and staff from three different NGOs: the Bungoma Red Cross, AMPATH, and ACE-Africa. Observation as a method gives more access to what people do [14]. Informal talks and the

participant approach during the programs allowed the investigator to get more insight into the issue, while creating trust among the study participants [14], whether health workers, volunteers, or those receiving services in the community. The 18 informants were recruited through purposeful sampling, and the interviewees were explicitly selected as they were likely to generate useful data [14]. When interviewing NGO staff and volunteers, community health workers, and public health officers, a discussion guide was used, setting the agenda and terms of the topic, and aiming to encourage the informants to speak openly and at length about the different themes [14]. Additional data from the period after the fieldwork and up to 2023 were also collected by reviewing both research reports and non-peer-reviewed documents, and through informal conversations on the jigger issue with health officers at the ministerial level. A search of the literature in Google, Google Scholar, Pubmed, and Hinari (WHO) was conducted in 2023, systematically searching titles, abstracts/articles, and content of the published literature, reports, and newspapers on the issue of jigger fleas and infestations in Kenya.

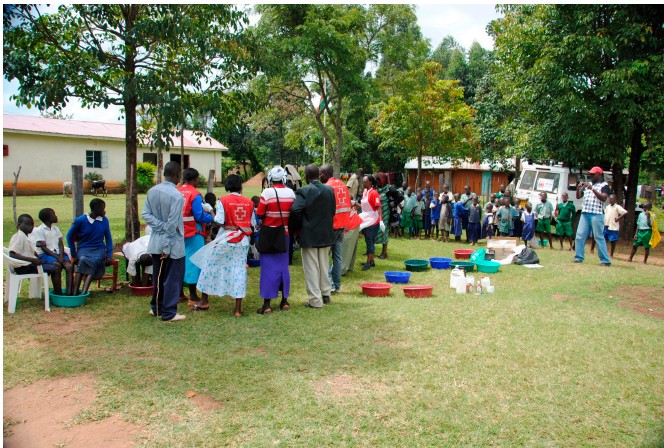

**Figure 2.** Jigger removal program, as conducted by the Bungoma Red Cross. Picture taken with permission by the first author.

All 18 in-depth interviews were conducted in English. Nine of the interviews were with NGO staff and volunteers attending the jigger removal clinics. Five of these interviews were with the NGO staff (2) and volunteers (3) of the Bungoma Red Cross, the NGO that facilitated, planned, and organized the jigger removal program. The remaining four interviews were with volunteers in two other local NGOs: AMPATH and ACE–Africa, who also contributed to the jigger removal program. Furthermore, nine interviews were conducted with government health workers, four with community health workers, and five with public health workers. The five interviews with NGO staff and volunteers who were facilitating the program (the Bungoma Red Cross), were conducted at the local Red Cross house. The remaining thirteen informants were recruited on the day of the jigger removal clinic in the respective villages. All the informants allowed the researcher to use a tape recorder during the interview. They were informed that they could withdraw from the interview at any time; however, no informants chose to do so. None of the study participants were given incentives.

The interviews and fieldnotes from observation and from informal talks were transcribed and transferred into the software OpenCode 3.2. The written material was discussed and analyzed through a thematic content analysis approach. The thematic analysis first divided the written material into meaning units (e.g., "*They* [the government] *deal with other problems such as HIV/AIDS and malaria*"). Meaning units were labeled with a code (e.g., prioritizing between different health issues). Codes were then divided into categories (e.g., "No Time to Follow Up" and the Reoccurrence of the Flea), and, finally, categories were divided into three recurrent themes (e.g., local perceptions on the government's responsibilities in combatting the plague) [14].

## 3. Results

The thematic analysis of the collected data resulted in three recurrent themes: (1) the NGO-driven jigger removal program as a (fragile) resource for local communities, (2) the need for more consistent collaboration between NGOs and public health services, and (3) the local perceptions of the government's responsibilities in combatting the plague.

### 3.1. The NGO-Driven Jigger Removal Program as a (Fragile) Resource for Local Communities
### 3.1.1. Appreciation of the NGO-Driven Program

During the informal talks and observations during fieldwork, affected people in the visited villages undoubtedly appreciated the program's importance. The effort by the NGO staff and volunteers, aiming to improve the quality of life of those affected, was highly appreciated. While attending the jigger removal campaigns in the villages and observing the volunteers and staff's efforts for those whom they served, there was no doubt that they wanted to help them and give them relief. The removal clinics were well-attended and popular in the villages we visited, with about one-thousand people in total attending the five observed events. In comparison, very few villagers attended any healthcare facilities to seek treatment for sandfleas. For instance, during an informal talk with a leader of a local dispensary, he estimated that he had: "*...less than 10 patients per month visiting the dispensary due to jigger infestation*". Health workers, teachers, members of the community, family members, and infected people expressed that they were very grateful that someone came to their villages to help with their neglected problems. They particularly appreciated that the services were free of charge. In addition, several of those infected by sandfleas explained that the mobile jigger removal program was particularly appreciated due to the distance between the other health facilities and the villages often being too long, and they often had difficulties walking.

The public health officers in the different divisions also appreciated the mobile jigger removal program driven by the NGOs. One explained that: "*They* [NGOs] *are doing a lot to this country. They are helping a lot*". Another community health worker elaborated that

> "*There are so many benefits [with the jigger removal clinic]. Because children are freed [from the jigger], they can play freely. You know that when they are infested, they cannot play, when they kick the ball, it is so painful. So, it is a very big benefit to a community as a whole...*"

The staff and volunteers at the NGOs often expressed the urge to help those affected, and one employee in the NGO that facilitated the jigger removal program stated that "*It is actually our mission to eradicate human suffering*". He gave an example:

> "*There was this man we treated in Malakisi. We were there one year ago [for a jigger' removal campaign]. I met him again three months ago. He was walking well now. He was happy to see us again and said that we had eradicated his suffering. He [said he] was about "to die".*

### 3.1.2. Local NGOs and Constant Insecurity

Even though virtually all informants genuinely appreciated the efforts from the NGO organizing the mobile jigger removal clinic, the program did face several challenges. For instance, all nine informants from the NGOs who participated in the in-depth interviews were concerned about the deficient economy and the lack of funding. From time to time, the NGOs expressed that they did not have money to buy chemicals to treat jigger infections, nor to cover the transportation costs to get to the other villages in the county.

Furthermore, looking at the issue of funding, the jigger removal clinics needed considerable amounts of labor to mobilize and conduct successful clinics. The program was dependent on voluntary work, as most of the work conducted by the NGOs was voluntary. One of the NGO employees explained that volunteers received an allowance of KES 300 (approx. 2 dollars) for a full day of work, and most of the volunteers either did not have a job, or they were students. If the economy did not allow for the allowance, it was difficult

to get enough people to complete the activities. Several of the NGO employees had similar concerns: "*At least we must give them* [the volunteers] *lunch allowances*" as "*most of the volunteers are lacking* [money for covering their] *basic needs themselves*". Even though the volunteers "*Are willing to help. . . and enjoy the work of volunteering*", an NGO employee stated that without the allowance or at least a free lunch, it was difficult to motivate volunteers to participate. One volunteer confirmed that "*We also need some sort of motivation. . . If we could get some financial support, not only allowances*".

### 3.2. The Need for More Consistent Collaboration between NGOs and Public Health Services

Even though the NGOs and the public health sector did appreciate each other's efforts, several of the informants also described that it could sometimes be challenging to communicate and cooperate. For instance, some health workers in the community called upon a more consistent mobilization and better communication with the NGO, as well as better coordination around the program. A public health officer, during one removal day in a village, explained that

> "*This morning I was not informed that I was supposed to assist you. . . I had to rush. We need a proper program so we know what date and what time so that we can organize all the patients and we can organize the community health workers to inform people. . . So that everybody knows that this day is a jigger removal day*".

Another day, a health worker supported this statement by explaining that

> "*They [the NGOs] need to involve the [local] health workers. Sometimes they have been doing things without informing health workers*".

On the other hand, some of the staff and volunteers in the NGOs and other informants explained that some health workers failed to inform the villagers and those in need of the services about when and where the jigger program would be conducted in their community. This statement was supported by several participants during informal talks on the removal days, who expressed that they were not informed about the upcoming program by the local healthcare workers. Several people also explained that they knew more people who needed the services and who would have attended had they been informed. An NGO employee elaborated on this:

> "*When we started [the mobile jigger removal clinic] they [the health workers in the community] did not want to collaborate. We were just going there on our own. We tried to talk to them, but they were not engaged. . . But there has been a change. I've been working with this program since it started, and I have seen a change [to the better]*".

### 3.2.1. Collaboration between NGOs

During the fieldwork, we met five different NGOs who worked within jigger eradication and volunteered in Bungoma County (including the Red Cross, AMPATH, and ACE–Africa). Several times during the jigger removal program, other NGOs were present to assist with the jigger eradication; however, they were not necessarily aware that other NGOs were already working with the same issue in the same area. In fact, some NGOs in Bungoma experienced that even the biggest and most well-funded NGOs in Kenya were reserved about cooperating and sharing experiences with others. This might have been due to some competition in providing services, or the desire among some NGOs to keep the field for themselves as something of a niche. An NGO employee in Bungoma talked about his experience with a bigger NGO working on jigger eradication located within Nairobi:

> "*They probably have more funds than us because they have been in the media a lot. I called them because I wanted to exchange information, but they did not call me back. Maybe they don't want anyone to interfere in their project*".

However, looking at the NGOs working locally in Bungoma, volunteers and staff that participated in the in-depth interviews explained that they had partnered and taught other NGOs regarding how to remove the jigger, confirming that they always welcomed other

NGOs to join their programs. One NGO volunteer explained, for instance, that "*we are working together, and we do invite each other to activities*".

Nonetheless, virtually all informants saw that it could be a useful advantage to co-operate; however, to be able to do so, several of the informants called upon the need for the coordination of the work of NGOs. Such coordination was required to cover all of the communities and places in need of services, as well as to avoid the duplication of services and wasted resources.

3.2.2. Poor Commitment from the Government: "*NGOs Are Doing It Now Because No One Else Is*"

Nevertheless, from the NGO workers' perspective, there was a perception that the government and public health authorities were not taking their responsibilities seriously, as one NGO employee explained that "*They* [the government] *relax. They have the mentality that NGOs are coming in to help*".

A public health officer expressed that she thought it was appropriate that NGOs were responsible for the jigger removal programs. She argued that the government did not have the necessary finances for public hygiene measures to fight jigger infections. However, the NGO volunteers and staff interpreted their role as coming only in addition to governmental health services. As one staff member expressed, "*We are just supposed to be an NGO helping*".

Several of the employees and volunteers in the three NGOs involved in the study explained that they had to step in and act regarding the jigger epidemic, due to the low engagement by the government, and they felt that they were required to solve this underprioritized problem alone. They explained that the government took no initiative to fight the scourge, and that public health servants in the county considered the fight against the sandflea problem exclusively as an NGO task. In fact, none of the eighteen informants in this study felt that the government was seriously involved in jigger eradication, and many claimed that interventions mainly happened in the central province of Kenya, near Nairobi, not in the rural areas of the western part of Kenya, such as Bungoma.

Another NGO member emphasized the need for several actors working together to fight the plague, but not least, he pointed out the health authorities' responsibilities:

> "*NGOs are doing it now because no one else is. . . But it should be an issue for all of us; NGOs, the government, politicians, the community. . . It is a public health issue, for the government, the Ministry of Health, and the public health officers. That is why we pay them and that is why we pay taxes. NGOs should just come and supplement it*".

He further elaborated on challenges such as the political actors hiding or not publicly recognizing sandflea infestations, due to stigma of poverty and underdevelopment that is associated with the scourge:

> "*I remember when we talked to members of the parliament in Bumula [one of the most infected areas in Bungoma]. We told them that we wanted to conduct a jigger removal program. They replied there was no problem with jigger in that area. The problem does not get highlighted because of the stigma and nobody accepts that the problem is there. So it all goes back to those infected. It is left for them to try to see how they can help themselves*".

The NGO staff member further explained that politicians do not want to admit that they have jigger fleas and infestations in their area, as this would be the same as saying "*we are poor*", potentially causing them to lose votes at the election.

Virtually all informants agreed that the government was not engaged and did not give attention to the jigger epidemic. An NGO employee stated that poor commitment from the government was due to "*poor leadership*". Getting the government involved in fighting this neglected plague was described as essential; not just at the local community level in Bungoma County, but also in Kenya, at the national level. Some of the public health authorities working in Bungoma felt that they were forgotten by the national authorities in Kenya. One public health officer, for instance, explained that he had a meeting with a jigger

committee from the government once in the community, aiming to coordinate services; however, *"after the meeting, I have never seen them again. They are not on the ground"*.

*3.3. Local Perceptions of the Government's Responsibilities in Combatting the Plague*
3.3.1. The Cycle of Poverty: "It [jigger] Is One of the Big Problems for Economic Growth in Our Country"

First, virtually all informants agreed that the main risk factor for jigger infection was poverty, and that it would not be possible to eradicate jigger infections as long as people lived in constant poverty in areas with high levels of jigger infestations. Observations, informal discussions, and interviews conducted in the current study all confirmed that those suffering from jigger infections were living in poor socio−economic conditions. Furthermore, those suffering from jigger infections were often hindered from working or attending school, which further raises the poverty level. A staff member from one NGO explained that

> *"Jigger are a major problem because it creates poverty. Kids that are infected with jigger are not attentive at school, which mean that they will not perform".*

Furthermore, the staff member elaborated,

> *"For now, our program is a temporary help. The permanent solution is when the issue of poverty is addressed when they build proper schools with floors. . .!"*

In fact, jiggers were reported to be an important factor in why people were unable to cover their basic needs in Bungoma County. Many people explained that they were not able to walk, grip, or focus when infected, which again meant that they could not work; for most of them, work implied farming and agriculture. A public health officer explained that

> *"Those infected cannot work. Especially in farms the productivity is low. It is one of the big problems for economic growth in our country".*

One NGO volunteer elaborated on his observation of one homestead infected by jigger fleas:

> *"Some houses we do follow up on, do not have beddings. It is very hard. You find them sleeping in poor conditions. And they do not have cemented floors".*

The need for follow-ups in vulnerable communities, and the discussion about who should be responsible for the follow-up in order to prevent any reoccurrence of the flea after treatment was another topic frequently raised by the informants.

3.3.2. "No Time to Follow Up" and the Reoccurrence of the Flea

Referring to the perceived lack of involvement by the public health sector, several informants raised the issue of deficient follow-up care as a missed opportunity. Such follow-up after treatment in the removal clinics would help avoid the recurrence of the sandflea. An NGO volunteer explained that

> *"We treat it, we leave, and we hope that the community health worker will follow up the cases. At times the number of community health workers is not sufficient. And they are not only dealing with jigger. They must deal with [many] other health issues".*

An NGO employee claimed that jigger infections are not prioritized and that "*They* [the government] *deal with other problems such as HIV/AIDS and malaria*".

Community health workers and NGO workers were both aware of the importance of follow-up after treatment in the prevention of the reoccurrence of the jigger flea. One NGO volunteer elaborates on the importance of visiting the living environment as a part of the follow-up with those affected:

> *"When we go to their homes, we see how the family is living. . . You find the problem and the cause of infestation. . . And you can speak with the family".*



However, they all explained that the lack of resources made it difficult to carry out the actual follow-up. One community health worker was frustrated and explained that "*the program is good, but it takes too long to reach all the people* [in need of the services]". Several of the community health workers in Bungoma did agree with the need for local follow-ups, and they saw it as their task to visit and re-visit the households; however, they too faced several challenges, such as a lack of drugs at the local dispensaries. One community health worker explained that "*...we do not have drugs available at all*". In addition, the individual treatment of those infected was described by the community health worker as too time-consuming, as one health worker was responsible for up to 125 households. As one community health worker explained: "*it can take a whole day to remove jigger from just one household. That is too time-consuming*". In addition, the community health workers felt that they had to focus on more severe diseases, such as HIV/AIDS. Health workers explained that, because of this, jigger infections tended to be ignored, as there was no time to follow up on every person who received services from the NGOs.

Virtually all informants agreed that it would be of great importance to establish household visits as a follow-up after treatment, combined with awareness raising on environmental sanitation and personal hygiene.

### 3.3.3. "We Are Using Our Own Knowledge": The Need for Guidelines and Awareness Raising

Virtually all informants explained the need for clear guidelines to fight jigger infections. A public health officer explained that "*We are using our own knowledge*". This was supported by other informants, like the leader of a local dispensary, for instance: "*We use our experience*". Neither those infected by jigger fleas, nor health workers who worked on jigger removal had any access to written information or brochures on how to prevent and treat jigger infestations, or how to avoid the recurrence of the flea after treatment. Regarding treatment, health workers reported differing measures to address jigger infections: some used alcohol or different disinfectants to kill the flea (such as Savlon or Dettol), and some still reported removing the flea with sharp needles, thus apparently not conforming to any common guidelines. Some community health workers indeed confirmed that they still used the traditional method of removing the fleas with pins or blades, even though this is painful, leaves open wounds, and increases the risk of secondary infections, or even of sharing diseases such as HIV/AIDS. As a district public health officer explained,

> "*There are no proper guidelines on the most effective way to treat jigger. There are different methods and thoughts about what is the most effective way to treat it*".

An NGO staff member supported this statement, and expanded on what several others had said:

> "*I call upon the need for proper research to be made by the Ministry of Health so that we can come up with proper guidelines. It would be of importance for both us and those affected*".

Hand in hand with the need for guidelines which are accessible for those treating jigger infection in the communities, the need for awareness raising was often mentioned. Several of the informants felt that those infected lacked significant knowledge of jigger fleas, and that household visits to those affected would be an opportunity to raise awareness of any preventive measures. These views were found among NGO workers, public health officers, and community health workers. One community health worker explained that

> "*Most of the people are not following instructions, we advise them to do this and that, and after a while they forget, and the jigger reoccur again. We need to keep educating them*".

They emphasized the need for health education in local communities and schools. They were convinced that to eradicate jigger fleas, the level of education had to be improved. As an employee in an NGO emphasized, "*The schools play a very important role in educating people about hygiene*".

Some of the informants also added that, in addition to awareness raising on personal hygiene and environmental sanitation, the fact that animals are carriers of jigger fleas might need to be more prevalent on the agenda. One public health officer explained that

*"At times when we look at the beddings, animals and cats and dogs are using the mattress during the daytime. As they scratch, the fleas remain in the blanket".*

However, in general, there seemed to be little focus among both public health workers and NGO staff and volunteers on the awareness of the fact that animals might be carriers and reservoirs of jigger fleas and infections. The issue of animals as reservoirs was rarely mentioned unless the topic was brought up by the researcher. Informal conversations with two veterinarians also indicated the lack of knowledge on the topic, as they explained that only small dogs could be infected by jigger fleas, and that jigger fleas were not a big issue among animals in Bungoma, thereby ignoring their key role as vectors for human infection. Thus, here too, there seemed to be a lack of collaboration across sectors, which is primarily a public responsibility.

3.3.4. The Need for Documenting Prevalence: "We Start with Those Who Call Us"

Both the volunteers and staff of NGOs, public health officers, and community health workers emphasized that it was necessary to get more knowledge on the prevalence of jigger infections in the villages of Bungoma County. At the time of the study, several NGO volunteers mentioned that, in areas where local people reported to them, they had frequent visits. However, they also suggested that there were other places where a contact person was not so accessible and thus needed more assistance. Several informants, therefore, called upon the need for mapping the situation in order to ensure proper sensitization. One NGO volunteer explained that

*"There is a need for a systematic survey to be done, about where the jigger are, in which areas, and in which populations. If we can do it with malaria, why can't we do it with jigger? . . . We should do a mapping of the villages and [after that] we can do a campaign in the areas most infested".*

As confirmed by NGO volunteers, it is difficult to know how and where to respond to the infestation when there are insufficient data, leaving much to chance. As an NGO volunteer explained,

*"We start with those who call us, and if we have time and resources after that, we can do assessment in other areas".*

During the fieldwork, we planned a jigger removal campaign in an area that was not heavily infected when one headteacher in the area had requested the NGO. Upon arrival, we were informed by community health workers that it would be better to go to the neighboring village, where they were struggling with jigger infestations. To sum up, mapping the prevalence of jigger fleas and infections seems essential to effectively combat the scourge, and, here again, the public responsibility for doing so was called upon locally.

## 4. Discussion

Virtually all the staff and volunteers of the different NGOs that were participating in the study explained that they were motivated to give relief and care to those affected by jigger infections in the local communities. A former study based on the same fieldwork in Bungoma [6] also found that those suffering from jigger infections highly appreciated the effort from the NGOs. The NGO-driven programs met people at their homeplace, which was essential, considering the widespread difficulties in walking for those affected, the transportation costs when travelling to health facilities, and the low expectations of getting necessary medication in local healthcare centers [6,25]. In addition, those affected were often afraid of being ridiculed and stigmatized [6,10,26]. Another factor that favored the NGO-driven jigger removal program was that those affected by jigger infections tended not to use mainstream health services for treatment [6,26]. Similarly, the national guidelines on

jigger prevention and control also recommend such outreach jigger removal programs, due to poor health-seeking behaviors [7]. Moreover, the NGO-driven activities are primarily based on the use of volunteers, and, due to the dependence on voluntary work, several of the informants felt that the program was fragile. The community coalition action theory (CCAT) emphasizes the need for training, defined roles, and positive results to be able to motivate volunteers, making them feel committed to the work they are doing towards jigger control and eradication. These factors might also be important in the recruitment of new volunteers [21]. For many of the volunteers who were poor and lacked paid jobs themselves, economic return was an important incentive. Even though most of the volunteers had an urge to help, this could be difficult when they had not fulfilled their own basic needs. Former studies support this, adding that governmental support for volunteers consisting of stipends, potential employment, supervision, and/or training could increase volunteers' motivation, and that inadequate remuneration and supplies could discourage the motivation of the volunteer [27]. Therefore, as supported by several of the informants, it might be precarious to depend on voluntary work when it comes to jigger control and prevention within the county. Most of the informants expressed the need for collaboration between NGOs and the local public health sector; however, the informants in Bungoma felt that they were left alone to solve the problem, and that public health servants in the county also considered it an NGO issue.

Several of the informants raised the issue that the work towards jigger control and prevention, especially within high endemic communities such as Bungoma [17], should be a national responsibility, tackled through collaborative problem solving and the coordination of the available resources in public and private sectors, according to the needs of the respective counties [18,19]. In the case of Kenya, coordination challenges between the national Ministry of Health, the county Department of Health, and NGOs are reported to negatively impact the efficiency of the Kenyan health system and compromise the health system's performance [28]. Furthermore, partnership between different sectors is indeed expressed as crucial to tackle the determinants of health and illness, something that is particularly important in the case of diseases and the conditions of poverty, as sandflea infestations affect mainly poor communities and counties. This requires leadership, a shared vision, clearly defined responsibilities, and good communication between the different stakeholders [19–21]. However, several of the informants indicated that the ministries are located far away from Bungoma County, both in terms of geography, but also in terms of their availability to assist those working within jigger control in rural areas. The national guidelines on jigger prevention and control, developed by The Ministry of Health, do indeed support the NGOs' request of governmental coordination. The guidelines emphasize that county and national governments should identify potential actors and stakeholders who perform outreach activities to ensure that all jigger prevention and control activities follow the guidelines [7]. Therefore, policy recommendations are important to maximize the effectiveness of community outreach programs [21]. In the policy recommendations, it is suggested that the Ministry of Health, through the Division of Environmental Health and in consultation with county departments for health and other stakeholders, should review and update the policy guidelines every third−fifth year in order to incorporate new research, progress, lessons learned, and changes within the disease epidemiology [7]. However, almost ten years after these guidelines were initially published, no review of the guidelines has been carried out.

The CCAT emphasizes the need for community capacity; in this case, involving the local community health workers in the district in the planning, mobilizing, and conducting of the jigger removal program [20,21]. In Kenya, the Alma Ata declaration on primary health care, stating that people have the right to participate in the planning of their health care, has contributed to an increased emphasis on decentralization in health care, and on the essential role of communities to the success of health and development programs. A study from Kilifi in Kenya showed that the engagement of the community through committees influenced target and priority setting, but the emphasis on national health indicators left

many local priorities unaddressed, and the final impact on budgets allocated at district and facility level was limited [29]. This suggests that engaging the community and local health workers in the planning process of jigger control is feasible, but also that there are challenges in ensuring that this local engagement feeds into implementation.

The Ministry of Public Health and Sanitation in Kenya suggests that targeting more than one disease will maximize resources [30]. Even though this was not specifically mentioned by any of the informants, the community health workers in Bungoma explained that they could be responsible for up to 125 households, and they needed to prioritize more severe diseases such as HIV/AIDS. Therefore, any follow-up after jigger removal programs was not an option; this might favor the need to combine different measures and services. For instance, in affected areas, combining the distribution of malaria nets, the spraying of houses, and the dissemination of information about preventive sanitation measures for jigger eradication, as well as preventive measures for other diseases, would indeed be more cost-effective and time-saving for health workers. As an example, another NGO working in Kenya, USAID, conducted a WASH program (water, sanitation, and hygiene). This program cuts across several sectors, focusing mainly on water supply, sanitation access, hygiene promotion, management, and environmental sustainability within rural areas [31]. As a part of the One Health approach, the WHO suggests that jigger removal should be included in such control programs, and should be incorporated in school-based interventions [17].

Better collaboration between different NGOs also emerged as fundamental for the eradication of jigger fleas and infections, according to different informants, as the number of NGOs that work on eradicating jigger infestations is increasing. Studies have indicated that several NGOs working on the same issue in the same area increase the chance for duplication [32], which was also suggested to be the case in our study. Former studies suggest that weak state capacity and leadership are primary reasons for this situation. The risk lies in the lack of mutual exchanges of information between different NGOs, thus resulting in insufficient cooperation and duplicating services [21,32]. To avoid duplicating services, presently in Kenya, a new NGO called HENNET (Health NGOs network) aims at coordinating and networking CSOs (community services) in the health sector. A hundred NGOs were registered as members of HENNET by 2020 [33]. All three NGOs that were participating in our study are listed as members of HENNET. However, when NGO-based informants expressed their concerns over duplicating services, HENNET was not mentioned by any of them. An estimation of the total number of NGOs that are working on jigger removals in Kenya could be found by using the NGOs Coordination Board [4]. When typing "jigger" and "tungiasis" in the search field, only two NGOs working on jigger removals came up: the Omonyakomu Community Development Organization and the Fighters of Poverty, Jigger and Drugs Rehabilitation Programme [4]. By using local networks in Kenya, as well as searching in Kenyan Newspapers, the following NGOs were confirmed to be working on jigger eradication in Kenya: Community Health Support (COHESU) [34], Helping Hands [35], Mannion Daniels' mission [36], Jigger Ahadi Trust [37], Step 30 [38], Red Cross [39], Ace Africa [40], AMREF [41], and Lund International Rotary Club [42]. The fact that more than 11 NGOs are found working towards jigger eradication, while the informants rarely knew or spoke about other organizations, might demonstrate the need for coordination between the NGOs working on jigger removals, including information on where the NGOs are located. The rapidly increasing number of international and national NGOs working with health issues in Kenya also increases the expectation that NGOs are following up on social services, which may to some degree have resulted in a dependency from the government towards NGOs [43]. This was confirmed by several of the informants in the current study, and the NGO staff and volunteers felt that they were given the responsibility of jigger prevention and eradication; however, this should not be their sole role. Nevertheless, as the government "*relaxed*", NGO workers and volunteers still felt obliged to respond and help all those suffering. It is stated that Western donor nations prioritizing donations to international NGOs over state entities contributes

to the growth of the NGO sector. This fuels the expectations among community members and policymakers that NGOs will assume leading roles [43]. All three NGOs interviewed in Bungoma collaborate with or are affiliated with an international NGO, receiving some form of international funding. This situation may underscore the issues previously mentioned regarding creating dependencies and expectations, thus obscuring the division of roles between public and private-not-for profit services [21].

Moreover, several of our informants, as well as some former research, suggest that the effect of the NGO-driven jigger removal program is temporary and short-term [21], unless the underlying issues of poverty and the reoccurrence of the flea are addressed and followed up at the community level [6,44,45]. Findings from the same fieldwork in Bungoma found that all members in four different households were reinfected again within three weeks of treatment [6], which favors the importance of primary preventive and environmental measures [20,46], awareness raising, and health education [20]. Green et al. (2019) emphasize that education and healthy public policies combined are important factors in achieving effective health promotion on issues such as jigger infections [19]. According to our informants, as a tool to combat the jigger menace, health education is an important yet marginalized factor. Health education might be crucial in enabling individuals and communities to gain control of their own health [14,20], and should include awareness raising and skill and knowledge development among those affected, combined with capacity building in the community, including the empowerment and education of school teachers, for example [19,21]. Indeed, even though the people exposed to the flea and health workers had knowledge on jigger infections and fleas, there was confusion around the preventive and treatment measures, as well as amongst the informants in the current study, which favors the need to include elements of community education [6,47–49] and evidence-based recommendations [19] within the jigger removal programs.

Furthermore, virtually all informants called upon the need for clear guidelines, as they had no access to any brochures or written information regarding jigger infections [6,48]. The fact that neither those infected [6] nor the public health workers or NGO staff and volunteers focused upon the potential role of animals as carriers of jigger indicates the need to inform and intervene on the role of both domestic and sylvatic animals as reservoirs for infection, in addition to education on environmental and personal hygiene [8,47,50]. This information is available in the national policy guidelines on the prevention and control of jigger infestations. The guidelines also emphasize the need for using various insect repellants as a prevention strategy [30], which informants in our study rarely mentioned. Furthermore, a study conducted after the national guidelines were written in 2014, indicates new and effective treatment methods, which are both non-toxic and non-expensive, and should be added as recommended modes to treat jigger infections and infestations in the national guidelines [51,52]. Thus, our study shows the need for more awareness-raising and training on effective treatment and eradication strategies among health workers at the grassroots level, whether government-employed or operating through NGOs. Additionally, this suggests that the existing guidelines lack updates on the research or best practices and fails to reach those actively working with jiggers in the country. To be "useful in achieving our vision of a jigger-free Kenya", as written by the Ministry of Health in the guidelines, the content should reach the public.

Finally, to know which areas are most in need of interventions, epidemiological data should be gathered [19,21]. In 2023, the CEO (chief executive officer) of the biggest NGO working on jigger eradication in Kenya, the Jigger Ahadi Trust, emphasized the need for government identification of infected areas in order to enable stakeholders to better plan for effective jigger eradication [53], as also supported by the national guidelines [7]. Thus, systematic data on the prevalence of sandfleas in Kenya are needed, as the data seem to be virtually non-existent in Kenya, the rest of East-Africa, and even worldwide [11,54]. All suggestions are in line with the health promotion principles that presuppose good governance for health where policymakers across all government departments make health a central line of government policy [20].

## 5. Limitations and Concluding Remarks

This qualitative case study aimed to address the perspectives of both government-employed health workers' and NGO staff and volunteers in Bungoma County. It sought to understand their views on the role division between the government and civil society organizations, as well as on the challenges and opportunities of jigger control and eradication.

A limitation of the study is that national stakeholders, such as the Jigger Ahadi Trust and the Ministry of Health, were not invited to participate. These national stakeholders were frequently mentioned by participants. Moreover, it would possibly be an advantage to interview more than nine NGO workers and nine governmental health workers in order to ensure saturation [14] and applicability outside the study setting [55]. However, the strength and novelty of this study is that it contributes to share the informants' rich, lived experiences and understanding of the control and eradication of this neglected affection [55]. Such perspectives on the role division of public and private not-for-profit organizations in combatting jigger infections have, to our knowledge, not been addressed in Kenya. Previous research has called upon the need for qualitative studies addressing the perspectives of those working with the plague; this is essential in the development and updating of guidelines and control programs [5,7,12,56]. Furthermore, the study sheds light on the need for comprehensive and coordinated actions at the policy level ensuring improved long-term outcomes for the affected communities [20,21].

**Author Contributions:** Conceptualization, Å.W.M. and G.V.d.B.; methodology, Å.W.M., G.V.d.B. and J.S.; software, Å.W.M. and G.V.d.B.; validation, Å.W.M. and G.V.d.B.; formal analysis, Å.W.M. and G.V.d.B.; investigation, Å.W.M.; resources, Å.W.M. and G.V.d.B.; data curation, Å.W.M., G.V.d.B. and J.S.; writing—original draft preparation, Å.W.M., G.V.d.B. and J.S.; writing—review and editing, Å.W.M., G.V.d.B. and J.S.; visualization, Å.W.M.; supervision, G.V.d.B. and J.S.; project administration, Å.W.M.; funding acquisition, Å.W.M. All authors have read and agreed to the published version of the manuscript.

**Funding:** The University of Bergen, Centre for International Health, partially funded Å.W.M.'s fieldwork in Kenya; Open Access Charge of the article is funded by NLA University College.

**Institutional Review Board Statement:** The study was conducted in accordance with the Declaration of Helsinki and approved by the Regional Committee for Medical and Health Research Ethics, Western Norway and by the Institutional Research and Ethics Committee (IREC) at Moi University in Eldoret, Kenya before data collection (FAN: IREC 000865).

**Informed Consent Statement:** Informed consent was obtained from all subjects involved in the study.

**Data Availability Statement:** All data can be accessed in OSF https://doi.org/10.17605/OSF.IO/FQ5KB. Contributor name: Åse Walle Mørkve.

**Acknowledgments:** We want to express our gratitude to all those who shared their stories during the Jigger removal program. Thank you to the staff and volunteers of Bungoma Red Cross, AMPATH and ACE-Africa, who shared their knowledge, aiming to help those suffering from jigger infections. To Amin Ali Sheikh, thank you for facilitating Åse Walle Mørkve's stay in Bungoma. To the research group "Intercultural meetings, Diversity and Inclusion" at NLA University College, thank you for input and support. To Alexander Bielicki, thank you for proofreading the article.

**Conflicts of Interest:** The authors have declared that no competing interests exist.

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
