# Peer review of "“We Are Just Supposed to Be an NGO Helping”: A Qualitative Case Study of Health Workers’ and Volunteers’ Perceptions of the Government and Civil Society’s Role in Fighting Jiggers in Bungoma County, Kenya"

_societies, doi:10.3390/soc14020028_

Round 1

Reviewer 1 Report

Comments and Suggestions for Authors

I like this paper.  It's basically descriptive in terms of your data collection.  But the last half of your discussion section gets into really interesting and important issues that are not really anticipated by the way you present data collection.  I think there ought to be some summarizing or theoretical presentation of that material towards the beginning of the paper.  The conclusions you present jump out at the reader in a surprising way and it's hard to keep track of all the summary statements you are making.  For me, these are the most important and compelling materials in the paper.

Comments on the Quality of English Language

There are minor errors, especially in the first pages of the paper.

Author Response

Cover letter for submission after minor revision

Title of manuscript: “We are just supposed to be an NGO helping”: A qualitative case study on health workers’ and volunteers’ perceptions of the government and civil society`s role in fighting jiggers in Bungoma County, Kenya.

Thank you for inviting us to resubmit the article.

We hereby submit a new manuscript.

As requested, a list with details of the revision to the manuscript and responses to the referee’s comment, is included. Please find attached the manuscript with tracked changes.

Reviewer

Comment from reviewer

Comment from author (line number refer to the manuscript with tracked changes)

1 and 2

Minor editing of English language required

Proof readning conducted by a native English speeker

1

1.     The last half of your discussion section gets into really interesting and important issues that are not really anticipated by the way you present data collection. I think there ought to be some summarizing or theoretical presentation of that material towards the beginning of the paper

Edited.

Line 81-98

Minor editing of discussion in line with the comment (587-697)

1

2.     The conclusions you present jump out at the reader in a surprising way and it's hard to keep track of all the summary statements you are making

Edited. Line 698-717

2

1.     Abstract – Lines 93 to 96 on page 3 mention different aims, which are  inconsistent from the aims on line 11 of the abstract, why?  Perhaps keep them consistent

Edited. Line 111-113

2

2.     Methods – The number of interviews are very low – normally 20 to 30 interviews are recommended by qualitative researchers. Is this study generalizable?

Edited. Line 709-711

2

3.     This is a minor issue but why is “the” in bold font online 137 page 4

Edited

2

4.     Another minor issue – but “however” has incorrect punctuation – should be semicolon and comma before and after the word, respectively.

Edited.

2

5.     Methods – lines 139-140 – the aims on lines 93 to 96 mention experiences of those affected by jiggers but the interviews d not reflect this

Edited. Line 111-113

2

6.     Methods - Any incentives given to the participants?

Edited. Line 201

2

7.     How was the analysis done? Was software used for coding e.g. NVivo?

Edited. Line 204.

2

8.     The research questions and aims are described but how were participants recruited into the study? What methods were employed?

Edited. Line 172-173

2

9.     What is novel here regarding this study?

Edited. Line 81-98

2

10.  What are the implications of the findings for future directions?  What does this research add to the current literature?

Edited.

Line 81-98

Line 703 – 705

Line 715-716

2

11.  What were the theoretical frameworks that informed this study?

Edited. Line 81-98

2

12.  Limitations – Good to see some of these were mentioned; what about the strengths of this study? Also, what about the low number of interviews?

Edited. Line 709-711

Reviewer 2 Report

Comments and Suggestions for Authors

This study used qualitative case study design that was conducted through three months of field work to shed light on health workers’ and volunteers’ perceptions of government and civil societies’ role in fighting sand flea infections in Kenya.  This study is important and reads well; however, I have some concerns/ recommendations for this paper that could improve it further:

1.     Abstract – Lines 93 to 96 on page 3 mention different aims, which are  inconsistent from the aims on line 11 of the abstract, why?  Perhaps keep them consistent

2.     Methods – The number of interviews are very low – normally 20 to 30 interviews are recommended by qualitative researchers. Is this study generalizable?

3.     This is a minor issue but why is “the” in bold font online 137 page 4

4.     Another minor issue – but “however” has incorrect punctuation – should be semicolon and comma before and after the word, respectively.

5.     Methods – lines 139-140 – the aims on lines 93 to 96 mention experiences of those affected by jiggers but the interviews d not reflect this.

6.     Methods - Any incentives given to the participants?

7.     How was the analysis done? Was software used for coding e.g. NVivo?

8.     The research questions and aims are described but how were participants recruited into the study? What methods were employed?

9.     What is novel here regarding this study?

10.  What are the implications of the findings for future directions?  What does this research add to the current literature?

11.  What were the theoretical frameworks that informed this study?

12.  Limitations – Good to see some of these were mentioned; what about the strengths of this study? Also, what about the low number of interviews?

 Thank you for the opportunity to review this manuscript.

Comments on the Quality of English Language

Line 167 "however” has incorrect punctuation – should be semicolon and comma before and after the word, respectively.

Author Response

(The authors gave the same response as above.)

Round 2

Reviewer 1 Report

Comments and Suggestions for Authors

This version is much improved and the broader analytic points are well integrated into the text.

Author Response

Dear reviewer, 

Please find attached the cover letter. 

Reviewer 2 Report

Comments and Suggestions for Authors

Peer Review for Societies

This study used qualitative case study design that was conducted through three months of field work to shed light on health workers’ and volunteers’ perceptions of government and civil society's role in fighting sand flea infections in Kenya.  This study is important and reads well, and a number of improvements were done. However, I still have some concerns/ recommendations for this paper that could improve it further:

1.     There is still no mention about the generalizability of this study. Is this study generalizable?

2.    What is novel here regarding this study? This has still not been addressed.

3.  What are the implications of the findings for future directions?  What does this research add to the current literature? – this is still missing.

4.  What were the theoretical frameworks that informed this study? – This is completely missing.

5.  Limitations – I still did not see any strengths of this study that were added? Also, there is a mention of an advantage of conducting more than 9 interviews with each type/category of participants, but not why.  How would this be beneficial? E.g. for the generalizability?

Thank you for the opportunity to review this manuscript.

Author Response

Dear Reviewer,

Please find attached the cover letter.

Round 3

Reviewer 2 Report

Comments and Suggestions for Authors

The authors have addressed all of my concerns. Thank you for contributing such an interesting study.